# High-resolution AFM structure of DNA G-wires in aqueous solution

Krishnashish Bose[1], Christopher J. Lech[1], Brahim Heddi[1,2] & Anh Tuân Phan[1]

We investigate the self-assembly of short pieces of the *Tetrahymena* telomeric DNA sequence d[$G_4T_2G_4$] in physiologically relevant aqueous solution using atomic force microscopy (AFM). Wire-like structures (G-wires) of 3.0 nm height with well-defined surface periodic features were observed. Analysis of high-resolution AFM images allowed their classification based on the periodicity of these features. A major species is identified with periodic features of 4.3 nm displaying left-handed ridges or zigzag features on the molecular surface. A minor species shows primarily left-handed periodic features of 2.2 nm. In addition to 4.3 and 2.2 nm ridges, background features with periodicity of 0.9 nm are also observed. Using molecular modeling and simulation, we identify a molecular structure that can explain both the periodicity and handedness of the major G-wire species. Our results demonstrate the potential structural diversity of G-wire formation and provide valuable insight into the structure of higher-order intermolecular G-quadruplexes. Our results also demonstrate how AFM can be combined with simulation to gain insight into biomolecular structure.

[1] Division of Physics and Applied Physics, School of Physical and Mathematical Sciences, Nanyang Technological University, Singapore 637371, Singapore. [2] Present address: Laboratoire de Biologie et de Pharmacologie Appliquée, CNRS, Ecole Normale Supérieure, Paris-Saclay, France. Correspondence and requests for materials should be addressed to A.T.P. (email: phantuan@ntu.edu.sg)

The ability of nucleic acids to self-assemble into functional macromolecules has implications in both biology and nanotechnology[1–3]. A G-tetrad can be formed by association of four guanines through Hoogsteen hydrogen bonding[4]. Several G-tetrad layers can stack upon each other to form G-quadruplex DNA or RNA[5–7]. It has been shown that G-quadruplex blocks can further stack to form higher-order structures[8–10]. The *Tetrahymena* telomeric sequence d[$G_4T_2G_4$] was one of the first reported sequences known to self-assemble into wire-like structures, termed as G-wires, as visualized by dry atomic force microscopy (AFM) imaging on mica[11].

G-wires have received much interest from the nanotechnology community for their applications in nano-electronics[12–15], nanosensors[16–18], and nanodevices[19–21]. It has been proposed that long-range charge transport in G-wires may allow their use as conductive nanowires[22]. Recently, experimental studies have reported conductive behavior in single G-wire molecules of lengths over 100 nm[13]. Despite the progress that has been made to understand the electronic properties of DNA G-wires, there is lingering uncertainty about how individual G-rich strands assemble and what structure they adopt. In previous theoretical work, we demonstrated that the conductive properties of G-tetrads can vary by orders of magnitude among different commonly adopted orientations[23]. Understanding the structure of G-wires is crucial for the controlled assembly of systems[24] with optimal nano-mechanical and electrical properties. Additionally, in biological systems, fragments of guanine-rich DNA and RNA might self-assemble to form G-wires and accumulate in the cell. Recently, it has been reported that G-quadruplex-based aggregates may be associated with the C9orf72 expansion-mediated amyotrophic lateral sclerosis (ALS) and frontotemporal dementia (FTD)[25].

Several short guanine-rich DNA sequences have been reported to form G-wires as observed by scanning probe microscopy[26–38] in dehydrated or partially hydrated environments. Here we report high-resolution AFM images of G-wires in an ionic aqueous environment and focus on the *Tetrahymena* telomeric DNA sequence d[$G_4T_2G_4$], which has been extensively studied in the context of G-wire formation. We used AFM to probe higher-order structures formed by the self-assembly of the oligonucleotide d[$G_4T_2G_4$] in aqueous solution. Using a direct cantilever excitation method, we obtained high-resolution images of DNA G-wires in aqueous solution. A large dataset of high-resolution AFM images enabled classification of G-wire diversity based on height, periodicity, and handedness of the surface features. We used molecular modeling and molecular dynamics (MD) simulations to generate atomic models of G-wires consisting of several strands of d[$G_4T_2G_4$]. Simulation of AFM images from these models provided an explanation for our experimental observations.

## Results

**Formation of high-order G-quadruplex structures**. To probe the formation of G-wires in solution, nuclear magnetic resonance (NMR) and circular dichroism (CD) spectroscopy were performed. A DNA sample of d[$G_4T_2G_4$] was dissolved in a buffer containing 30 mM potassium phosphate (pH 7), 70 mM KCl, and 10 mM MgCl$_2$. NMR spectra showed several sharp imino proton peaks and a broad hump at 10–12 p.p.m. (Supplementary Fig. 1), characteristic of G-quadruplex formation. Upon annealing the sample (heating and slowly cooling), the sharp imino protons disappeared with only a broad hump remaining. These observations indicate the possible formation of multiple and/or higher-order G-quadruplex-based structures, which were detectable by solution NMR spectroscopy.

CD spectra of d[$G_4T_2G_4$] showed a positive peak at 260 nm and a trough at 240 nm (Supplementary Fig. 1), consistent with the formation of parallel G-quadruplexes. At 95 °C, the intensity of the 260 nm peak was reduced by only ~ 30% as compared with that at 25 °C (Supplementary Fig. 2), indicating that a significant fraction of the parallel G-quadruplex structures formed by d[$G_4T_2G_4$], remained stable at high temperature (95 °C).

**Observation of G-wire and duplex DNA by AFM in aqueous solution**. The main goal of this study was to understand the formation of the higher-order structures of d[$G_4T_2G_4$] in aqueous solution by using AFM. We first used AFM to image these structures in air. After sample annealing, DNA G-wires of lengths of up to 200 nm and heights of 2.3 ± 0.2 nm (for 1085 molecules) were observed (Supplementary Fig. 3). These AFM observations in air are consistent with those reported in the literature[26].

To prepare G-wires for solution AFM study, 50 μM DNA solution was annealed in 20 mM HEPES buffer (pH 7) containing 50 mM KCl and 10 mM MgCl$_2$. For comparison purposes, we used a 320 bp duplex DNA as a reference, which was added to the d[$G_4T_2G_4$] solution just before immobilization on the mica substrate. Using direct photothermal excitation, we could image these structures without significantly perturbing them while scanning. AFM images in solution (Fig. 1 and Supplementary Fig. 4) revealed relatively straight wires of length 7–100 nm (Supplementary Fig. 5) with a height of 3.0 ± 0.3 nm (Supplementary Figs. 6, 7) and curved molecules of length 109 ± 4 nm (Supplementary Fig. 8) with an average height of 2.1 ± 0.4 nm (Supplementary Figs. 6, 7). Isolated blobs with length of < 7 nm

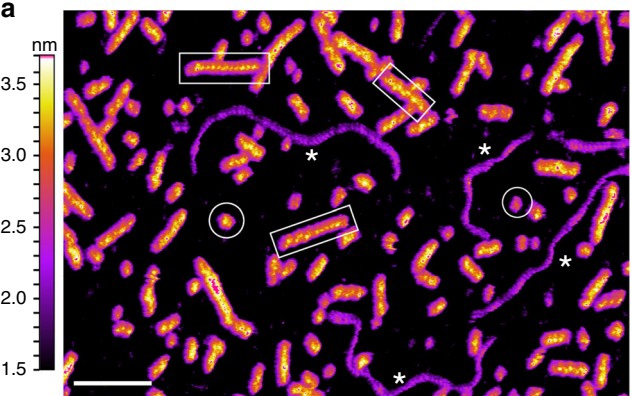

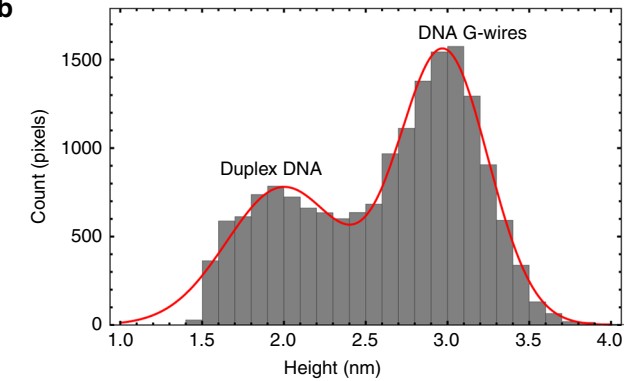

**Fig. 1** Visualization of DNA G-wires in aqueous solution. **a** AFM height image of G-wires and duplex DNA. Selected duplex DNA are indicated by asterisks (*). Selected G-wires are boxed. Objects with length < 7 nm and height > 2 nm were identified as isolated G-quadruplex blobs (circled). The length scale (white bar) is 30 nm. **b** Histogram of pixel height taken along the central spline of duplex DNA and G-wires with a length >10 nm from the above AFM image (full image)

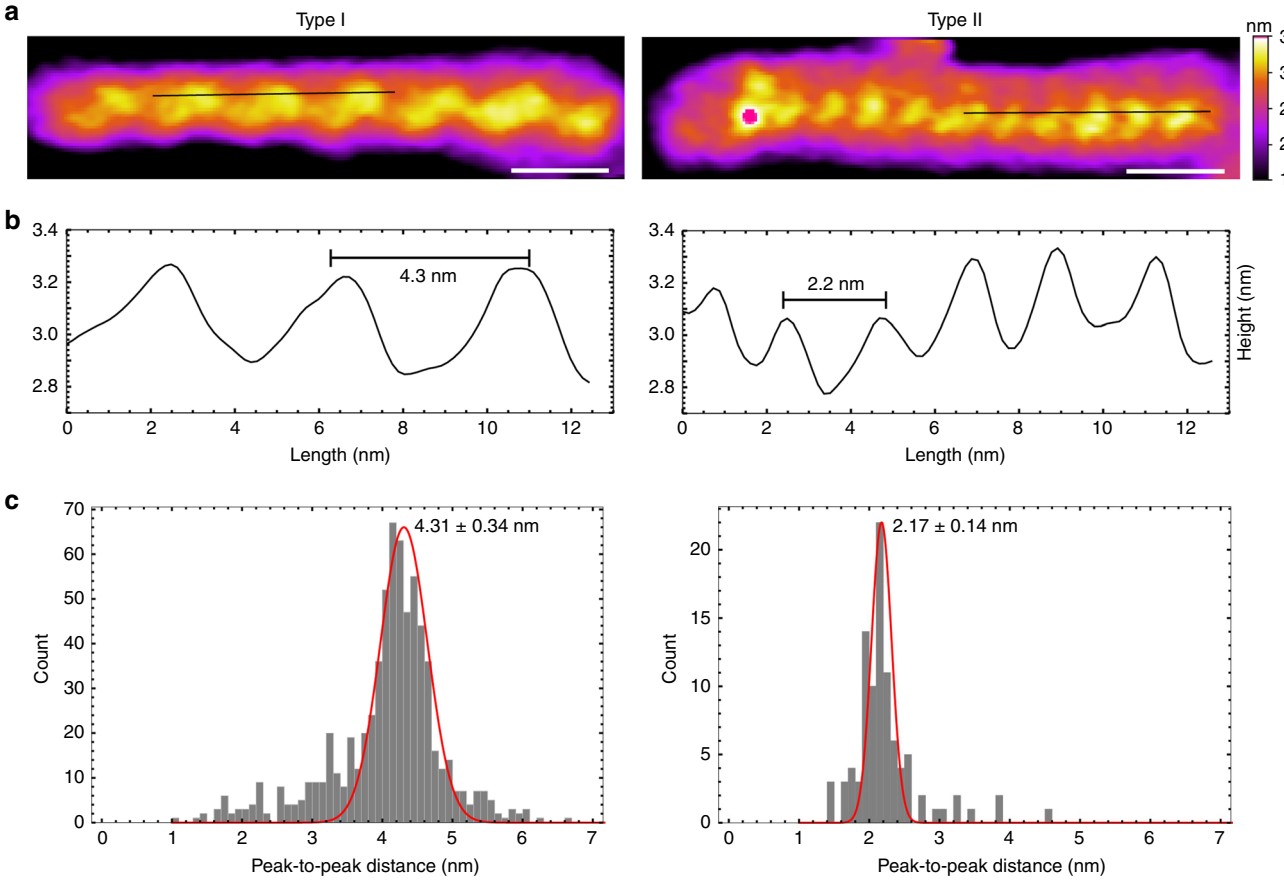

**Fig. 2** Major types of G-wires. **a** AFM images and **b** height profiles (under the black lines) of Type I and Type II G-wires showing periodic ridges (peaks) separated by 4.3 and 2.2 nm, respectively. The length scale (white bar) is 5 nm. **c** Histograms of peak-to-peak distances fitted with Gaussian indicate periodicity of 4.31 ± 0.34 and 2.17 ± 0.14 nm for Type I (100 molecules) and Type II (10 molecules) G-wires, respectively

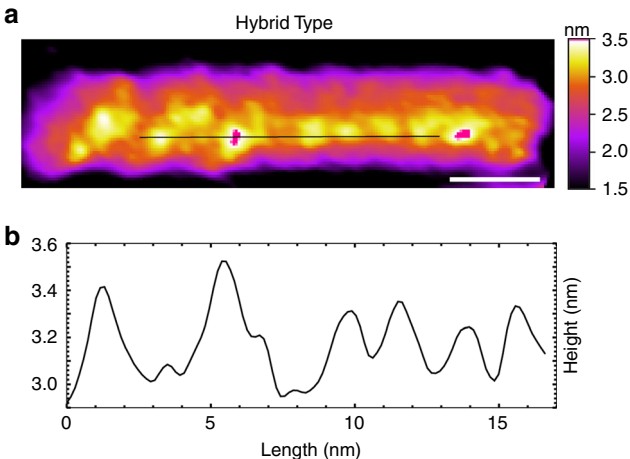

**Fig. 3** Hybrid-Type G-wire. **a** AFM image and **b** height profile of a Hybrid-Type G-wire demonstrating distinct regions of different periodicities. The length scale (white bar) is 5 nm

and height > 2 nm were also observed, which were likely to be isolated DNA G-quadruplexes.

We assigned the curved molecules as reference duplex DNA based on the following: (1) they demonstrate periodic features in agreement with the minor and major grooves of duplex DNA as reported in previous AFM studies[39,40] (Supplementary Fig. 9); (2) the height observed is similar to that previously obtained by solution AFM at low imaging forces[39,40]; and (3) the observed lengths (Supplementary Fig. 8) are consistent with the expected

length of 109 nm for a 320 bp B-DNA duplex with 0.34 nm separation per base pair.

The species of relatively straight wires are assigned to be G-wires based on the following: (1) they are 0.9 ± 0.03 nm higher than the duplex DNA species (Supplementary Fig. 7), which is in-line with previous reports[26] of d[G4T2G4] G-wires being consistently higher then duplex DNA; (2) the wires are straighter compared with duplex B-DNA, as previously reported[26]; (3) their lengths are diverse (Supplementary Fig. 5) and fall within the range of G-wire lengths reported in the literature[30]. Collectively, these results suggest that we were able to observe G-wires immobilized on the mica surface in an aqueous environment.

**Diversity in G-wire surface features**. The resolution of our AFM images allowed observation of periodic features in many of the deposited G-wires. Height profiles along the length of G-wires revealed notable differences in the surface features of these molecules. From 74 high-resolution AFM images, each containing 30–100 molecules obtained on 3 different days from 3 different sample batches (of 4, 28, and 42 images, respectively), the periodic surface features of G-wires were visually inspected, selected molecules were cropped, and their height profiles were extracted. In general, G-wires were observed to exhibit 4.3 or 2.2 nm periodicity, which we classified as Type I or Type II, respectively (Fig. 2).

Type I G-wires were observed in all the high-resolution images as the major species, of which 100 were individually analyzed (see Methods) displaying surface features with a periodicity of 4.3 ± 0.3 nm (Supplementary Fig. 10). Over the course of our study,

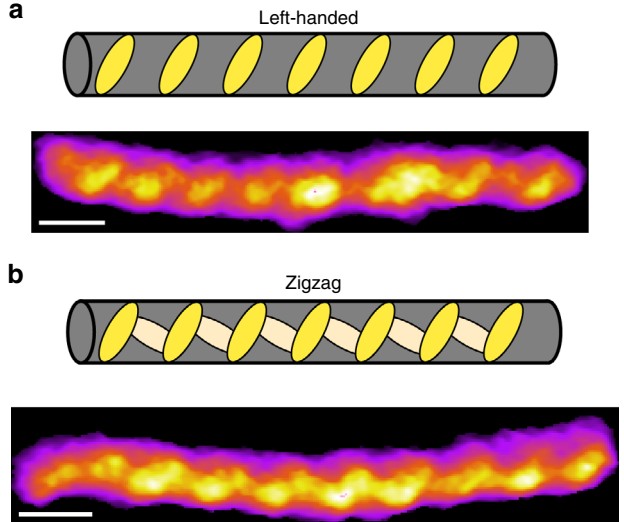

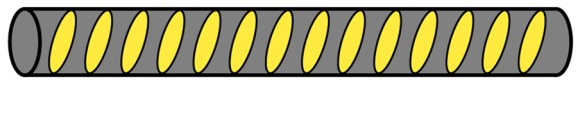

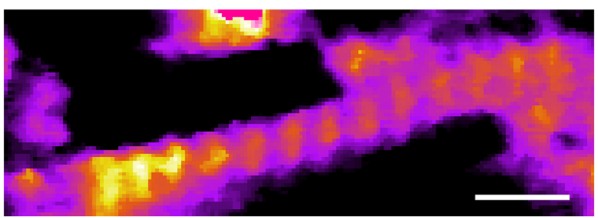

**Fig. 5** Left-handed Type II G-wire. AFM image and illustrative schematic of a left-handed Type II G-wire. The length scale (white bar) is 5 nm

**Fig. 4** Subtypes of the Type I G-wires. AFM images and illustrative schematics of the **a** left-handed and **b** zigzag subtype of the Type I G-wire. The length scale (white bar) is 5 nm

only 15 high-resolution images of Type II G-wires were found. Despite this low occurrence, Type II G-wires were observed across multiple images for two different sample batches (made 4 months apart) and imaged with two different types of AFM probes (Biolever mini and FastScan D). Ten Type II G-wires were analyzed displaying surface features with a periodicity of $2.2 \pm 0.1$ nm (Supplementary Fig. 11).

We also observed G-wires with a mixture of periodic features (Fig. 3), which we term as Hybrid-Type G-wires. These G-wires consisted of distinct segments demonstrating 4 and 2 nm periodic surface features, respectively. However, only three such G-wires (Supplementary Fig. 12) were observed over the course of our study.

Furthermore, subtle differences were observed in the surface features of G-wires within each type. In Type I G-wires, two kinds of progression of the periodic features were observed (Fig. 4). The left-handed subtype was characterized by distinct ridges that progress in a left-handed manner along the length of the molecule. The zigzag subtype demonstrates similar dominant left-handed ridges with additional features that connect these ridges to trace out a zigzag pattern on the G-wire surface.

Within the Type II G-wire, differences in the surface features were also observed (Fig. 5 and Supplementary Figs. 13, 14). The scarce observation of Type II G-wires and the less pronounced tilt angle of the surface features led us to cautiously define two separate subtypes. Out of the 15 Type II G-wires observed, 13 of them contained distinct ridges progressing along the length of the molecule in a left-handed fashion. However, two observed G-wires could be classified as a right-handed subtype, which appeared to contain ridges progressing in a right-handed manner (Supplementary Fig. 13).

Moreover, even smaller periodicities were observed occasionally, although the magnitudes of these features were small. Images of 51 Type I G-wires revealed periodic surface features with $0.9 \pm 0.2$ nm separation at 88 sections identified manually (Fig. 6 and Supplementary Fig. 14).

**Structural model building**. We set out to investigate how the structure and arrangement of d[$G_4T_2G_4$] strands could give rise to the diverse patterns observed with AFM. Despite the overwhelming possibilities in which strands could arrange themselves

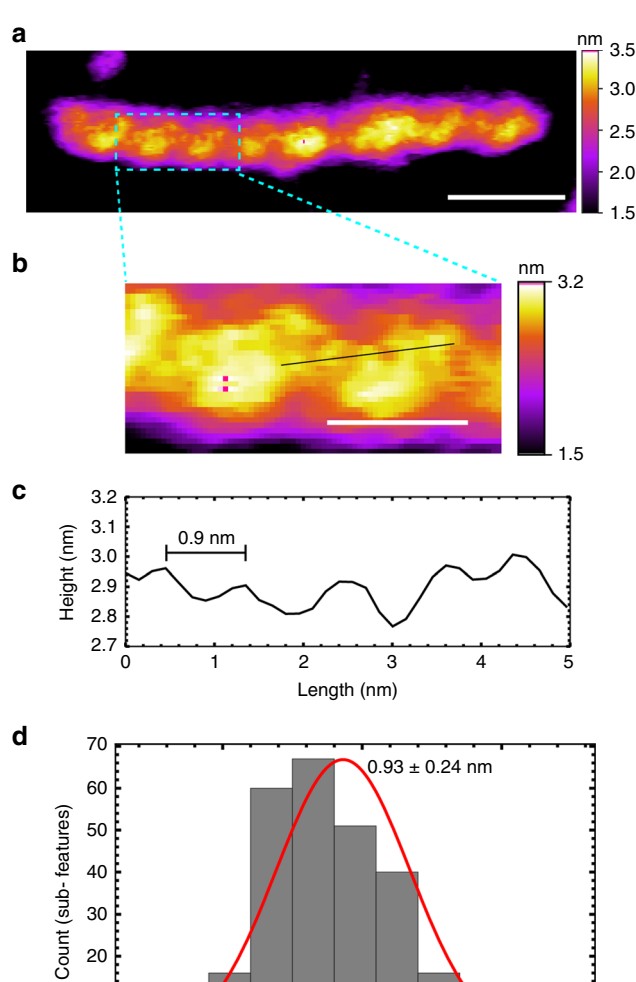

**Fig. 6** Observation of 0.9 nm periodicity. **a** AFM image of a Type I G-wire. The length scale (white bar) is 10 nm. **b** A zoomed portion of the above image revealing periodic right-handed sub-features. The length scale (white bar) is 4 nm. **c** The height profile from the selected section. **d** Periodicity histogram from 88 sections of 51 Type I G-wires revealed peak-to-peak distance of $0.93 \pm 0.24$ nm

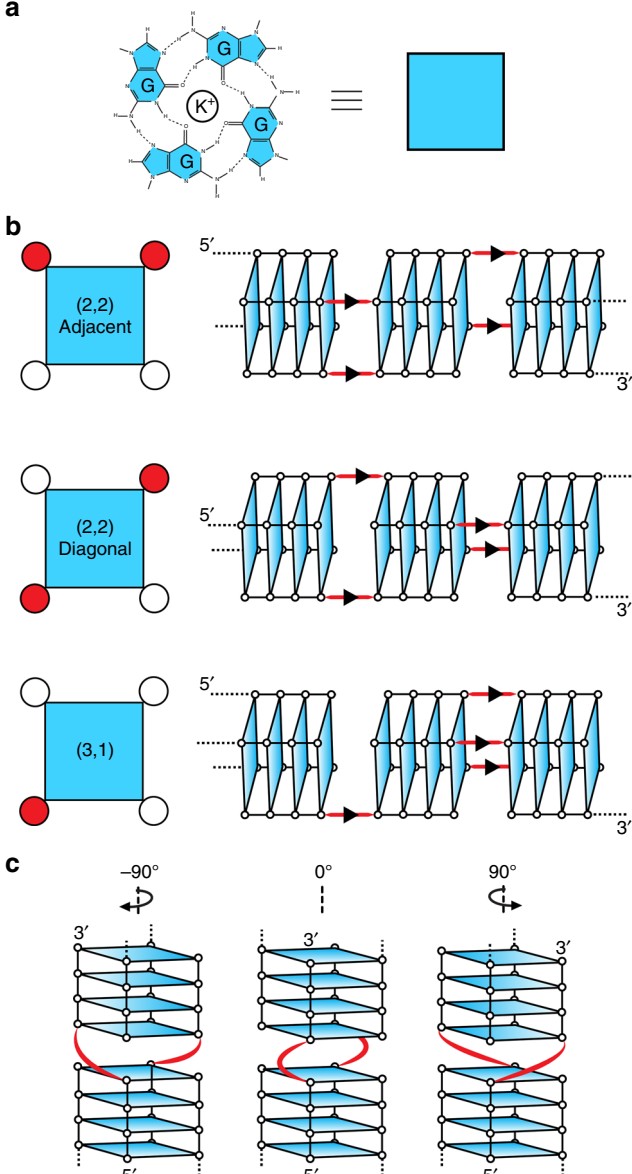

**Fig. 7** Structural models of G-wires. **a** A G-tetrad is represented as a square for simplifying schematic representation of various G-wire models. **b** Three slipped-strand models: "(2,2) Adjacent", "(2,2) Diagonal" and "(3,1)" models. Left: top view of a G-tetrad; slipped strand indicated as a red dot. Right: side view of a G-wire fragment containing three G-tetrad blocks; G-tetrads are represented as planes. **c** Rotamers of the "(2,2) Diagonal" slipped-strand model: $-90°$, $0°$ and $+90°$ rotation occurring at the TT thymine bulge (red). Rotation angle nomenclature is defined by the relative rotation of the top G-tetrad block with respect to the bottom G-tetrad block and the $5' \rightarrow 3'$-strand directionality as indicated

including the possibilties of partial pairing between G-tracts, we focused our investigation on variations of the slipped-strand arrangement originally proposed by Marsh et al.[41], in which a G-tract (GGGG) of a d[$G_4T_2G_4$] strand fully pairs with G-tracts of other d[$G_4T_2G_4$] strands (Fig. 7). In this manner, interlocked four-layered G-quadruplexes are formed from four strands coming together in parallel and recruiting other strands in a sticky-end fashion, eventually forming long wires with a G-tetrad core. A parallel arrangement of the strands would be consistent with the CD profile showing a positive peak at 260 nm.

Within these constraints, three different slipped-strand arrangements are possible (Fig. 7b). In "(3,1)" arrangement, a single strand is slipped with respect to the others. In "(2,2) Adjacent" arrangement, two neighboring strands are slipped. Finally, in the "(2,2) Diagonal" arrangement, strands located on opposite corners of the G-tetrad are slipped.

Adding further complexity, within each of these three arrangements, rotational variations may occur at the interface where one four-layered G-quadruplex block meets another (Fig. 7c). Similar diversity at the stacking interface of G-quadruplex blocks has been previously reported[42]. In the slipped strand model, interface junctions are connected by a TT linker, which could allow for G-quadruplex blocks to adopt three different rotational conformers (Fig. 7c). We define these different rotamers as $-90°$, $0°$, and $+90°$ based on their relative rotation from a natural backbone progression in the $5' \rightarrow 3'$-strand direction. Rotation by $180°$ was not investigated as the TT linker would be unfavorably stretched. Computational approaches were used to investigate the three slipped-strand structures (Fig. 7b), each with three rotamers (Fig. 7c), leading to nine models (Supplementary Fig. 15).

To understand whether any of the slipped-strand models could give rise to the experimentally observed AFM images, we built nine different G-wire models of 11 nm length, comprising 16 single strands of d[$G_4T_2G_4$]. Longer wires were constructed by rotation/translation of the 11 nm models (see Methods) and relaxed using MD simulation. After 1 ns of unrestrained MD simulation, all nine models maintained their global structure as well as the rotamer conformation at the interface junctions. Furthermore, the stacking of guanine bases at the junction interface adopted either a 'Partial 5/6-ring' configuration or a slightly offset rotation of ~45° previously demonstrated to be energetically favorable for unconstrained G-tetrad stacking[42].

Out of the models explored, only the -90° rotamer of the "(2,2) Diagonal" arrangement exhibited good agreement with the experimentally observed AFM data for Type I G-wires (Fig. 8); both in surface periodicity (4.3 nm, 0.9 nm) and handedness (left-handed and/or zigzag). Interestingly, two different orientations of this model could explain both subtypes of Type I G-wires (left-handed and zigzag). None of the models explored appeared to reproduce the 2.2 nm periodic features of Type II G-wires, suggesting that an unexplored type of strand arrangement might be involved. We note that some other simulated models produce alternative periodic surface patterns (Supplementary Fig. 16).

## Discussion
In 1994, Marsh et al.[41] proposed a model for G-wires formed by d[$G_4T_2G_4$], in which two adjacent strands slip and overlap to form four-layer interlocked G-quadruplexes, which we refer to as the "(2,2) Adjacent" arrangement. Since 1994, several G-wire models have been proposed, which vary in sequence and structure[28,31,32,37,43–45]. In the current work, we demonstrate that a slipped-strand model can give rise to the surface features observed in experimental AFM data. Importantly, only the "(2,2) Diagonal" arrangement in a $-90°$ rotamer configuration yields the 4.3 nm periodic features observed experimentally. We therefore propose that the $-90°$ rotamer of the "(2,2) Diagonal" configuration is a dominant species formed by d[$G_4T_2G_4$] in our experiments. It is interesting that the thymines of the d[$G_4T_2G_4$] sequence are seemingly responsible for the surface features observed in AFM. Even more interesting is how these thymine bulges protrude out to produce features that are cyclical and generally left-handed in nature. This finding might help to explain the observation of 3.5 nm periodical features previously

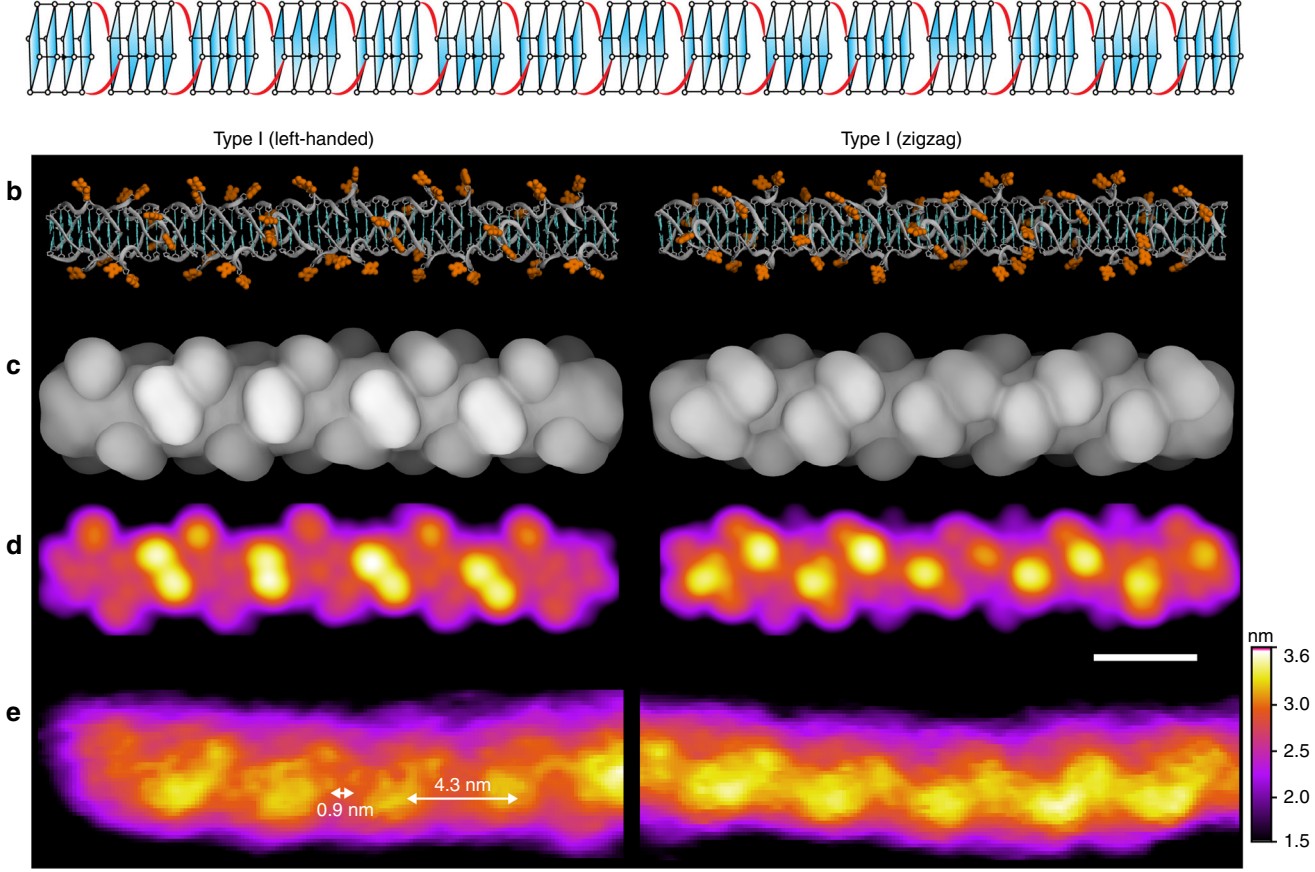

**Fig. 8** Comparison of proposed model with AFM observation of *Type I* G-wires. **a** Schematic representation of − 90° rotamer of the "(2,2) Diagonal" model containing 15 G-tetrad blocks, corresponding to a length of ~ 20 nm. **b** The G-wire structural model for the − 90° rotamer of the "(2,2) Diagonal" arrangement in two different molecular orientations. Thymines are shown in orange. **c** Surface representation of the above structures, where the grayscale intensity is related to the distance from the plane over which the molecule is lying. Atomic radii for Thymines have been altered for visualization purposes. **d** AFM images generated by simulated scanning with a paraboloid AFM tip. The length scale (white bar) is 4 nm. **e** Experimentally observed AFM images of Type I G-wires showing two types of periodic surface features

reported for scanning tunneling microscopy (STM) images of poly-G molecules[46], which might be formed by protruded guanine bulges or loops[47].

The 0.9 nm periodic features observed in some of the high-resolution AFM images of G-wires most likely correspond to the periodicity of the phosphate backbone running in a right-handed progression behind the thymine bulges (Fig. 8).

The observation of Hybrid-Type G-wires is intriguing as it suggests that Type I and Type II G-wires have some degree of topological similarity or compatibility regarding strand orientation and stacking interface that allow the two different structures to combine into one.

It is inherently difficult to correlate AFM observations of G-wires in solution to their molecular structure. Differences in AFM surface features could arise not only from structural differences in G-wires, but also from variability in G-wire orientation on the mica surface and from variation in G-wire imaging parameters. For example, the difference between left-handed and zigzag subtypes of Type I molecules is quite subjective for some observed molecules. Furthermore, it is not without notice that the 2.2 nm periodicity of Type II G-wires is about a half of the 4.3 nm features of Type I. This is consistent with a model of stacked G-tetrads having molecular periodicities, which are several times the distance between individual G-tetrad layers. More broadly, there is a notable percentage of G-wires in AFM images that

exhibit no observable periodic features. Such surface features could arise from other strand arrangements, which aren't inherently periodic in nature, or arise from molecules that are poorly attached to the mica surface. Despite these difficulties, the repeatable observations of Type I molecules and the unique agreement with the − 90° rotamer of the "(2,2) Diagonal" slipped-strand model suggests that the bulk of a given sample adopt these structures in solution.

As per our model (Fig. 8), a G-wire of ~ 20 nm length contains 15 G4 blocks composed of 28 full and 4 half strands of d[$G_4T_2G_4$]. It has a molecular weight of ~ 100 kDa, which should be detectable as a broad hump in solution proton NMR spectrum. Significantly longer wires, as observed by AFM, may be beyond the detection limit of solution NMR spectroscopy, and an increase in the population of such long wires after annealing (Supplementary Fig. 3) may explain the observed reduction in the integral area under the proton NMR spectrum (Supplementary Fig. 1).

Our proposed molecular model not only conforms well with the high-resolution topography images, but also mechanical property maps of loss tangent, dissipation energy and stiffness as obtained with bimodal imaging in amplitude modulation–frequency modulation (AM–FM) mode (Supplementary Figs. 17, 18). It was observed that the peaks in height correlated directly with peaks in loss tangent and dissipation energy. In each of the three channels, a periodic pattern of 4 nm was observed. The second harmonic

frequency channel, which is directly related to stiffness, also revealed periodic features, but not with well-defined 4 nm periodicity. The additional peaks might be indicative of additional structural details with variable stiffness. Interestingly, G-wires exhibited higher values of loss tangent, dissipation energy and stiffness compared to duplex DNA (Supplementary Fig. 19), suggesting that G-wires are stiffer than duplex DNA.

Understanding the structure of G-wires in physiological conditions is important considering the implications of G-rich aggregation in disease pathogenesis. The results presented herein have particular implications for the structure of the (GGGGCC)$_n$ repeat expansion in the C9orf72 gene, which is associated with neurodegenerative diseases like ALS and FTD[48,49]. We postulate that r(GGGGCC)$_n$ repeats in C9orf72 messenger RNA could potentially aggregate to form RNA G-wires in a manner similar to DNA G-wires formed by the d[GGGGTTGGGG] sequence as reported in this work.

AFM images of G-wires formed by the d[G$_4$T$_2$G$_4$] sequence in aqueous solution reveal several periodic features (4.3, 2.2, and 0.9 nm) and indicate the formation of polymorphic DNA G-wires. We built structural models based on the slip-strand arrangement of G-wire formation and compared simulated and experimental AFM data to derive structural insights. We demonstrate that a particular rotamer of the "(2,2) Diagonal" arrangement is capable of explaining the majority of G-wire features observed in AFM images. The diversity and structural basis for G-wire formation reported in this work may have significant implications in nanotechnology involving DNA self-assembly and also possible relevance in biology.

## Methods

**DNA sample preparation**. DNA oligonucleotide d[G$_4$T$_2$G$_4$], used for AFM experiments, was purchased from IDT Singapore (250 nmol, lyophilized, standard desalting). Samples were dissolved in 10 mM potassium phosphate (KPi) buffer (pH 7) at a concentration of 200 μM and stored at − 30 °C. Duplex DNA of 320 bp was amplified using PCR from an *Escherichia coli* pHT582 plasmid[50]. For G-wire preparation, a 50 μL solution (Sample Solution) was prepared containing 50 μM d[G$_4$T$_2$G$_4$], 50 mM KCl, 20 mM HEPES, and 10 mM MgCl$_2$. The sample solution was annealed by heating in boiling water bath and cooled slowly to room temperature. It was stored at room temperature for several weeks. The DNA sample for NMR experiments was synthesized on an ABI synthesizer using products from Glen Research and purified using Glen Research's protocol, followed by dialysis and lyophilization.

**DNA immobilization**. In our hands, existing protocols for duplex DNA immobilization in solution were insufficient to properly immobilize DNA G-wires. We developed two key steps that enabled us to visualize G-wires in solution. (1) Wash the freshly cleaved mica surface with DI water before sample loading. We suspect this reduces potassium content and other impurities from the mica surface. (2) Incubate G-wires (30–50 ng/μL) with 10–20 mM NiCl$_2$ just before deposition on mica. In our experience, this allows for better immobilization of G-wires on the mica surface allowing better resolution in AFM images (Supplementary Fig. 20). We expect this new protocol to be useful for solution AFM imaging of G-quadruplexes in general. With this approach, even 0.2 mM NiCl$_2$ was found to be sufficient for immobilization of duplex B-DNA in aqueous solution (Supplementary Fig. 21).

**Atomic force microscopy**. Experiments were performed using a Cypher ES AFM purchased from Asylum Research (Oxford Instruments). Solution AFM images were obtained in amplitude modulated ac mode (commonly known as tapping mode) using FastScan-D probes (Bruker) having a cantilever of 16 μm length and 4 μm width. Biolever mini AFM probes (from Asylum Research) were used for a few initial trials but did not allow reproducible high-resolution imaging of G-wires. Experiments were performed at 10 °C (via sample stage cooling), at which the FastScan-D probes had a resonant frequency of 90–100 kHz, which is slightly lower than the frequency at room temperature. A pulsed blue laser (BlueDrive) with ∼ 5 μm circular spot size was focused at the base of the cantilever for photothermal excitation[51]. For detecting cantilever deflections, a superluminescent diode (SLD) laser with 3 × 9 μm spot size was used. Cantilever amplitudes of < 1 nm with imaging speeds of 10,000 pixels/s were used (only) for high-resolution imaging.

**Mechanical property measurement**. The Cypher ES AFM can use multiple feedback systems (lock-in amplifiers) allowing excitation and monitoring of multiple eigenmodes of the AFM cantilever simultaneously. AM–FM mode was used for determining dissipation energy, stiffness, and loss tangent of DNA. The resonant frequencies ($f_1$, $f_2$) and quality factors of the first and second harmonic of the cantilever was determined from the thermal noise spectra. Prior to imaging, the deflection sensitivity and force constant ($k_1$) of the cantilever was measured. The force constant of the second harmonic is automatically determined by the software using the relation: $k_2 = k_1*(f_1/f_2)$. The free amplitude for second harmonic was set to 25 mV, whereas for the first harmonic it was initially set to 250 mV. During imaging, the amplitude setpoint of the 1$^{st}$ harmonic was gradually reduced until intricate surface features on DNA showed up. The second harmonic amplitude was left untouched and it was ensured that it was not reduced to < 20 mV. It was also ensured that the first harmonic amplitude is much higher than the second harmonic amplitude. The imaging parameters were suitably chosen to image in repulsive mode (phase < 90°). The amplitude and frequency gains for second harmonic were increased to several tens of thousands.

**Image processing**. AFM images were first flattened using Asylum Research's AFM software, by masking all DNA and G-wires using a suitable height threshold (usually ∼ 1 nm), and then applying first-order flattening to the unmasked region. Gwyddion (open-source software) was used for all subsequent processing of the flattened AFM images[52]. For periodicity analysis from high-resolution images, a two-dimensional (2D) fast Fourier transform (FFT) filter was applied to get rid of high frequency periodic noise along the slow scan axis followed by a median filter of 3 pixel-width (Supplementary Fig. 22) and Gaussian filter with $\sigma = 2$. All images in the paper have the same pixel size of 0.15 nm, except for the Type II G-wire in Fig. 5, which was acquired using biolever mini at much slower imaging speed. The filtering described above, facilitated the splining process which otherwise led to irreducible branching of the spline. No FFT filtering was done for the images used for height analysis. No Gaussian filtering was used on the images from which the 0.9 nm sub-features were extracted. Height profiles for the 0.9 nm sub-features were extracted by drawing lines in Gwyddion on selected sections of the molecule. The peak-to-peak distances were determined from the height profiles by a code written in Mathematica.

**Periodicity analysis**. A program was written in Wolfram Mathematica (ver. 10) for systematic analysis of periodicity of DNA G-wires from 9 AFM images with 122 detected molecules of length > 25 nm, median height > 1.6 nm, and pixel count > 500. Spline of these molecules were obtained using Mathematica's default morphological thinning method, applying Otsu's threshold after Gaussian smoothing ($\sigma = 3$) of the cropped image and pruning ( = 10). Any improper spline was corrected by adjusting the intensity threshold in the lowest possible increment/ decrement over Otsu's threshold. The spline of the molecule was justified as proper if it met the following criteria: (1) no branches; (2) parallel to the longitudinal axis of the molecule; (3) similar length as the entire molecule; and (4) unaffected by any neighboring objects. The spline was then fitted with a suitable polynomial of order varying from 1 to 10 based on the straightness of the G-wire. For most cases, a fourth order polynomial offered an excellent fit to both straight and curvy splines. The height profile was then extracted from the original cropped image, using coordinates from the fitted spline.

Height profiles along the length of the molecule were extracted from the spline coordinates. Several height profiles were obtained by translating parallel to the spline along the length of the molecule. The height profile, which showed the least standard deviation in periodicity, was selected for periodicity analysis. Periodicity was measured as the peak-to-peak distance in height profiles. For peak detection, Mathematica's default method was used after smoothening of the height profile with Gaussian filter ($\sigma = 4$).

**Height analysis**. Even after proper flattening of the AFM images using Asylum Research's AFM software, the estimated "zero" height of the image was slightly inaccurate ( < 0.5 nm). To overcome this problem, the "zero" height was determined from the full height histogram of the images as the value of height corresponding to the highest peak in the histogram. This method works well for images where the molecules are sparsely located so that the background occupies the maximum area of the image. After proper assignment of "zero height," the images were thresholded using Mathematica's in-built Otsu's algorithm. Objects with pixel count of less than 1000 pixels were removed. A Gaussian filter ($\sigma = 3$) was then applied to the images for obtaining the spline of all objects. The coordinates from these splines were traced back to the original image, which was median filtered with 3-pixel width (no FFT or Gaussian filtering). The height values in the original image corresponding to the coordinates of the spline were plotted as a histogram.

**Model building**. Small 5 nm models consisting of 10–12 DNA strands of d[G$_4$T$_2$G$_4$] were assembled using XPLOR[53]. Standard distance constraints were used that exist between any two G-tetrads of an arbitrary parallel-stranded G-quadruplex crystal structure. The constraint files were generated using Dev-C++. Minimal energy structures were identified and manually translated/rotated symmetrically using Pymol to create longer models. Fourteen full and four truncated

ssDNA strands generated G-wires of ~ 11 nm length. Central $K^+$ ions were added (33 in total) in between tetrads and on the exterior of terminal exposed G-tetrads.

**MD simulation**. The long G-wire models were simulated using MD approaches for relaxation in explicit solvent. The input files were prepared using xleap (Amber-Tools15) and simulated using Amber 10[54] with the ff99 DNA force field[55]. Models underwent unrestrained energy minimization of 1000 steps of steepest decent followed by 9000 steps of conjugant gradient method to remove clashes generated by manual model building. More $K^+$ ions were added to neutralize the system. The structures were then solvated with water (truncated octahedron, TIP3P).

Systems underwent an initial restrained minimization involving 500 steps of steepest decent followed by 500 steps of conjugated gradient while maintaining 25 kcal mol$^{-1}$ Å$^{-2}$ harmonic potential restraints on solute atoms. Systems were next heated from 100 to 300 K over 10 ps under constant volume while maintaining solute constraints. Systems further underwent a gradual relaxation of constraints in which systems underwent cycles of minimization and restrained dynamics with constraints of 5, 4, 3, 2, 1, and 0.5 kcal mol$^{-1}$ Å$^{-2}$. Finally, simulations were run unrestrained at 1 atm and 300 K for a duration of 1 ns. Molecules were visualized using VMD and Pymol. Molecular structures were averaged over the 1 ns unrestrained simulation and used for AFM simulations.

**AFM simulation**. Simulated AFM images were generated by a custom script written in MATLAB. In brief, AFM height images are produced for an input set of molecular coordinates by scanning a virtual paraboloid AFM tip (with a latus rectum 'a' of 0.16 nm$^{-1}$) over the molecule. The equation of the paraboloid is given by: $z = a(x - x_0)^2 + a(y - y_0)^2 + z_0$, where $(x_0, y_0, z_0)$ is the position of the vertex of the paraboloid in nanometers, whereas the molecule to be imaged is lying on the $XY$ lane at $z = 0$. In simpler words, at a height of 4 nm from the vertex of the AFM tip, a circular cross-section of the paraboloid AFM tip has a radius of 10 nm. The AFM simulation was also done with $a = 0.08$ nm$^{-1}$, but it was deemed to yield images that were unrealistically high resolution (Supplementary Fig. 23). The closest distance of approach of the simulated AFM tip without colliding with the molecule was registered as the contact height and stored as intensity in the simulated AFM image. Orientation of the molecule was varied as − 90°, 0°, and 90°, thus generating a total of 27 simulated AFM images. The 0° orientation of the molecule is arbitrary, as generated by the MD simulation average of 1 ns. The images were saved as grayscale image and opened with the Gwyddion program for AFM color rendering, Gaussian smoothening and extraction of height profiles.

Particularly for the − 90° rotamer of the "(2,2) Diagonal" model, longer structures were simulated which were built using the structure averaged over 1 ns of the MD simulation (Fig. 8). In this case, a suitable orientation of the molecule was chosen so that it could be phased/aligned well with the AFM image.

**Circular dichroism**. CD spectra were recorded at 25 °C using a JASCO-815 spectropolarimeter with a 1 cm path length quartz cuvette containing a solution volume of 500 μL. Strand DNA concentration was ~ 10 μM.

**NMR spectroscopy**. NMR experiments were performed on a 600 MHz Bruker spectrometer at 25 °C. DNA concentration was ~ 0.5 mM. Solution contained 30 mM potassium phosphate (pH 7), 70 mM KCl, 10 mM MgCl$_2$, and 10% D$_2$O.

**Data availability**. The data that support the findings of this study are available from the corresponding author upon request.

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

## Acknowledgements

This research was supported by Singapore National Research Foundation Investigatorship (NRF-NRFI2017-09), Ministry of Education Academic Research Fund Tier 3 (MOE2012-T3-1-001) and Tier 2 (MOE2015-T2-1-092), and grants from Nanyang Technological University to A.T.P.

## Author contributions

K.B. designed and performed all experiments and analysis under the supervision of A.T. P. C.J.L. assisted K.B. with writing of the paper and computational analysis. B.H. helped with structural calculations and MD simulations. All authors were involved in active discussion on the project and paper. A.T.P and K.B. finalized the paper.

## Additional information

**Competing interests:** The authors declare no competing interests.

