## [Peer Review File · Nature Communications]

Reviewers' comments:

Reviewer #1 (Remarks to the Author):

This manuscript describes a high-resolution AFM study of DNA G-wires in aqueous solution. Its key result is a successful comparison of the AFM data to one of the proposed models for G-wire structure. My knowledge of the G-wire literature is somewhat limited, so I refer to other reviewers for comments on the interest and relevance of this work in that context, while noting a general interest in the application of AFM as a tool for structure determination (which is rather exceptional); and restrict the rest of my report to a technical evaluation of the manuscript.

Overall, this manuscript reports an elegant use of state-of-the-art AFM in solution to study G-wires. Compared to papers that use a whole battery of technically elaborated methods to study a problem, this manuscript is pleasantly straightforward. It reports AFM data of G-wires with implicit control by imaging double-stranded DNA at high (double-helix) resolution (Fig. 1), describes and analyses the measured topography on G-wires (Figs 2-6), discusses different proposed models for G-wire structure (Fig. 7), and finds a good match with one of these models. Overall, the study is carried out well and is technically sound. There are however quite a few smaller technical issues that will need to be addressed, as listed below.

- End of Introduction, “the use of ultra-short AFM cantilevers”: The used levers are not “ultra-short” compared to commercially available cantilevers nowadays. It is better to simply quantify the cantilever length in microns rather than using subjective terms for describing them, i.e., refrain from the term ultra-short throughout the manuscript.
- In the Results section, “Observation of G-wire and duplex DNA ...”, height values are sometimes given with errors and sometimes with an approximate value. It would be better to state ALL height values in the manuscript as value +/- confidence interval, and at the first instance indicate what the notation implies, e.g. “2.3 +/- 0.x nm (mean +/- standard deviation, n=...)”.
- Structures don't “float away while scanning”, but can be perturbed by the scanning probe.
- The authors claim sub-nanometre lateral resolution, but only (though impressively) resolve features 1 nm apart. Hence the claim of sub-nanometre lateral resolution is not justified.
- The description of the data in Fig. 1 is somewhat confusing, since referring to “curved wires” for the DNA duplexes, which puts the reader on the wrong track since earlier on, “wires” imply G-wires. Maybe “curved lines” would be better?
- Fig. 1: For completeness, it would be good to also include a complete height distribution of the

whole image, or at least a distribution in such a way that the reference height (zero, the substrate?) is clearly defined. E.g., extend the central spline somewhat beyond the ends of the structures such that the histograms contain a peak for the substrate height.

- In the subsection “Diversity in G-wire structure features”, the authors are unnecessarily vague with numbers. E.g. “over 100 such wires” - why not give the exact number; “very few high-resolution images of Type II G-wires” - how many, and in how many independent experiments; “for at least two different sample batches” - for how many sample batches exactly, two, three, four? This is important to allow the reader an unbiased assessment of the importance of the Type II features.

- Fig. 5: Unlike the other main Figs, Fig. 5 does not contain a color scale bar. Since the data in Fig. 5 appear to refer to only few cases, it is not so clear how these fit into the overall story, and how reproducible these observations are (e.g., with respect to tip dependence). This Fig. may therefore be referred to the SI.

- Fig. 6: This Fig. should be clearer about where in A the height profiles in B were recorded. In addition, the caption in B refers to red and black curves: There are no red curves.

- In the “structural model building”, the authors’ choice for the slip-strand arrangement appears rather arbitrary. Can this choice be motivated in more objective terms?

- Fig. 7 is rather hard to penetrate. It would benefit from redesign and at least clearer annotation within the panels.

- In Fig. 8, it would also be beneficial for clarity to include a first line of panels that shows the relevant models in similar schematic form as in Fig. 7. In addition, the labelling appears incorrect: E is missing in the caption, and C does not appear experimental AFM data to me. For clarity “excellent fit and particularly the periodicity and handedness” should be marked in the Fig. panels.

- In the Discussion, it is unclear how the cited STM results (“3.5 nm periodicity”) relate the observations in this manuscript.

- In the Methods, the Image Processing subsection lacks clarity: 2D-FTT filtering can easily lead to image artifacts, which makes it very important to specify what the exact filtering conditions are, including cut-offs in terms of lateral distances (ideally, FTT filtering should be avoided altogether). The Gaussian filter is only specified in terms of pixels, not in terms of the (more relevant) lateral distances.

- Similarly, the Analysis with Mathematic contains descriptions that are too vague at present. What do “suitable parameters” refer to, which objective criteria were used to determine if parameters were “optimized”.

- Similarly, the AFM simulation subsection refers to “subjectively chosen” tip radii. How were these chosen, how was “goodness” identified? And can the radii be given? That would be good to have an idea of the sharpness of the probes with which such AFM data can be obtained. The authors also mention the use of Gwyddion for “further analysis”: What analysis does this refer to? And how do the authors determine a “suitable orientation” for alignment with the AFM image.

- Fig. S3C the histograms overlap to the extent that they obscure each other (e.g., the “Before annealing” is not visible. And the legend missed the dark red annotation.
- Fig. S7 is hard to read and difficult to understand. The caption should be more explicit about the procedure used to determine the histograms, and about the reference height (zero, see also comment related to Fig. 1 above).
- The data in Fig. S10 are not convincing and may be omitted from the manuscript.

Reviewer #2 (Remarks to the Author):

The work provides insight into structural features of the G-wires formed by the DNA sequence d(G4T2G4) in potassium/magnesium solutions. It is indeed one more study that provides insight into the mechanism of self-assembly and structure of G-wires. However, it is the first providing AFM images of G-wires in aqueous environment. Like most G-wires studied it contains a parallel stranded stem, but the resolution of the AFM images allows for left-handed features very likely due to thymine loops- I agree. The stem is still right-handed as the dichroic signal shows. Herein lies the novelty.

I believe that the manuscript can be accepted provided the issues below are addressed satisfactorily.

The authors make a broad claim in the Abstract: "However, to date, the mechanism of their assembly and the structure of G-wires remain unclear". By implication the authors should have addressed this issue. However, the claim is not backed up with any state-of-the-art related work in the Introduction. This should not be ambiguous.

A few issues have to be addressed:

The claim for right-handed features in the G-wire is not compelling. Page 7 lines 156-7. The authors state “Two observed G-wires could be classified as a “Right-Handed” subtype as it appears to contain ridges that progress in a right-handed manner.” The evidence for this is shown in Figure 5B. The image is not quite credible. Please compare contrast selection in Fig 8C (Type I). The authors should provide a more compelling image.

Page 13/14 The authors state “Furthermore, it is not without notice that the 2.1 nm periodicity of Type II G-wires is half of the 4.2 nm features of Type I” So, now what’s the implication?

Figure 7 Where it states “(E-F) Rotamers of the slipped-strand model.” It should be “(E-F) Rotamers of the (2,2) diagonal slipped-strand model.”

Either include references as evidence for the statement “Similar diversity at the stacking interface of G-quadruplex blocks has been previously reported” or take this out of the manuscript.

In Figure 8 “Atomic radii for Thymines and have bene altered for visualization purposes.”
Correct to “...been altered”

Responses to Reviewers:

Referee 1

► This manuscript describes a high-resolution AFM study of DNA G-wires in aqueous solution. Its key result is a successful comparison of the AFM data to one of the proposed models for G-wire structure. My knowledge of the G-wire literature is somewhat limited, so I refer to other reviewers for comments on the interest and relevance of this work in that context, while noting a general interest in the application of AFM as a tool for structure determination (which is rather exceptional); and restrict the rest of my report to a technical evaluation of the manuscript.

Overall, this manuscript reports an elegant use of state-of-the-art AFM in solution to study G-wires. Compared to papers that use a whole battery of technically elaborated methods to study a problem, this manuscript is pleasantly straightforward. It reports AFM data of G-wires with implicit control by imaging double-stranded DNA at high (double-helix) resolution (Fig. 1), describes and analyses the measured topography on G-wires (Figs 2-6), discusses different proposed models for G-wire structure (Fig. 7), and finds a good match with one of these models. Overall, the study is carried out well and is technically sound. There are however quite a few smaller technical issues that will need to be addressed, as listed below.

Answer: We would like to thank the referee for his/her appreciation of our work and for helpful comments to improve our paper.

- End of Introduction, “the use of ultra-short AFM cantilevers”: The used levers are not “ultra-short” compared to commercially available cantilevers nowadays. It is better to simply quantify the cantilever length in microns rather than using subjective terms for describing them, i.e., refrain from the term ultra-short throughout the manuscript.

Answer: Following the referee’s suggestion, we have removed the use of the term “ultra-short AFM cantilevers” throughout the manuscript and described the geometry of the cantilevers on page 18.

- In the Results section, “Observation of G-wire and duplex DNA ...”, height values are sometimes given with errors and sometimes with an approximate value. It would be better to state ALL height values in the manuscript as value +/- confidence interval, and at the first instance indicate what the notation implies, e.g. “2.3 +/- 0.x nm (mean +/- standard deviation, n=...)”.

Answer: We now show the mean and standard deviation for the reported values as suggested by the referee.

- Structures don’t “float away while scanning”, but can be perturbed by the scanning probe.

Answer: The statement has been now revised (page 5).

- The authors claim sub-nanometre lateral resolution, but only (though impressively) resolve features 1 nm apart. Hence the claim of sub-nanometre lateral resolution is not justified.

Answer: The claim has been now removed.

- The description of the data in Fig. 1 is somewhat confusing, since referring to “curved wires” for the DNA duplexes, which puts the reader on the wrong track since earlier on, “wires” imply G-wires. Maybe “curved lines” would be better?

Answer: We have revised this to “curved molecules”

- Fig. 1: For completeness, it would be good to also include a complete height distribution of the whole image, or at least a distribution in such a way that the reference height (zero, the substrate?) is clearly defined. E.g., extend the central spline somewhat beyond the ends of the structures such that the histograms contain a peak for the substrate height.

Answer: Based on the referees' suggestion, we have added new figures in the Supporting information (Figs S6 and S7).

- In the subsection "Diversity in G-wire structure features", the authors are unnecessarily vague with numbers. E.g. "over 100 such wires" - why not give the exact number; "very few high-resolution images of Type II G-wires" - how many, and in how many independent experiments; "for at least two different sample batches" - for how many sample batches exactly, two, three, four? This is important to allow the reader an unbiased assessment of the importance of the Type II features.

Answer: The specific numbers of molecules analyzed have been mentioned in the paper as suggested by the referee. Furthermore, quantitative analysis of the data has been presented as histograms in updated Figs 2 and 6.

- Fig. 5: Unlike the other main Figs, Fig. 5 does not contain a color scale bar. Since the data in Fig. 5 appear to refer to only few cases, it is not so clear how these fit into the overall story, and how reproducible these observations are (e.g., with respect to tip dependence). This Fig. may therefore be referred to the SI.

Answer: Fig 5B has been now moved to the SI.

- Fig. 6: This Fig. should be clearer about where in A the height profiles in B were recorded. In addition, the caption in B refers to red and black curves: There are no red curves.

Answer: Fig 6 has been now revised with more quantitative analysis presented as histogram.

- In the "structural model building", the authors' choice for the slip-strand arrangement appears rather arbitrary. Can this choice be motivated in more objective terms?

Answer: Rationale for the choice of slipped strand arrangement model has been now discussed on page 11.

- Fig. 7 is rather hard to penetrate. It would benefit from redesign and at least clearer annotation within the panels.

Answer: Figure 7 has been revised to make it more easily understandable.

- In Fig. 8, it would also be beneficial for clarity to include a first line of panels that shows the relevant models in similar schematic form as in Fig. 7. In addition, the labelling appears incorrect: E is missing in the caption, and C does not appear experimental AFM data to me. For clarity "excellent fit and particularly the periodicity and handedness" should be marked in the Fig. panels.

Answer: Fig 8 has been revised as suggested by the referee with schematics shown and distances clearly labeled. The caption has been updated.

- In the Discussion, it is unclear how the cited STM results ("3.5 nm periodicity") relate the observations in this manuscript.

Answer: This has been now discussed (page 15).

- In the Methods, the Image Processing subsection lacks clarity: 2D-FTT filtering can easily lead to image artifacts, which makes it very important to specify what the exact filtering conditions are, including cut-offs in terms of lateral distances (ideally, FTT filtering should be avoided altogether). The Gaussian filter is only specified in terms of pixels, not in terms of the (more relevant) lateral distances.

Answer: This section has been revised to give more specific information (page 19).

- Similarly, the Analysis with Mathematic contains descriptions that are too vague at present. What do "suitable parameters" refer to, which objective criteria were used to determine if parameters were "optimized".

Answer: This section has been revised to give more specific information and clarifications (page 19).

- Similarly, the AFM simulation subsection refers to "subjectively chosen" tip radii. How were these chosen, how was "goodness" identified? And can the radii be given? That would be good to have an idea of the sharpness of the probes with which such AFM data can be obtained. The authors also mention the use of Gwyddion for "further analysis": What analysis does this refer to? And how do the authors determine a "suitable orientation" for alignment with the AFM image.

Answer: This section has been revised to give more specific information and clarifications (page 21-22).

- Fig. S3C the histograms overlap to the extent that they obscure each other (e.g., the "Before annealing" is not visible. And the legend missed the dark red annotation.

Answer: The caption of the figure has been updated for clarification.

- Fig. S7 is hard to read and difficult to understand. The caption should be more explicit about the procedure used to determine the histograms, and about the reference height (zero, see also comment related to Fig. 1 above).

Answer: New Supplementary Figures S6 and S7 have been included in SI.

- The data in Fig. S10 are not convincing and may be omitted from the manuscript.

Answer: This figure has been removed. Quantitative analysis of this pattern has been now presented as histogram in Fig. 6.

Referee 2

► The work provides insight into structural features of the G-wires formed by the DNA sequence d(G4T2G4) in potassium/magnesium solutions. It is indeed one more study that provides insight into the mechanism of self-assembly and structure of G-wires. However, it is the first providing AFM images of G-wires in aqueous environment. Like most G-wires studied it contains a parallel stranded stem, but the resolution of the AFM images allows for left-handed features very likely due to thymine loops- I agree. The stem is still right-handed as the dichroic signal shows. Herein lies the novelty.

I believe that the manuscript can be accepted provided the issues below are addressed satisfactorily.

Answer: We would like to thank the referee for his/her appreciation of our work and for helpful comments to improve our paper.

► The authors make a broad claim in the Abstract: "However, to date, the mechanism of their assembly and the structure of G-wires remain unclear". By implication the authors should have addressed this issue. However, the claim is not backed up with any state-of-the-art related work in the Introduction. This should not be ambiguous.

Answer: We have now re-written the Abstract to address the referee's concern.

► A few issues have to be addressed:

The claim for right-handed features in the G-wire is not compelling. Page 7 lines 156-7. The authors state "Two observed G-wires could be classified as a "Right-Handed" subtype as it appears to contain ridges that progress in a right-handed manner." The evidence for this is shown in Figure 5B. The image is not quite credible. Please compare contrast selection in Fig 8C (Type I). The authors should provide a more compelling image.

Answer: We have now moved this image to the SI (also suggested by referee #1) given the uncertainty of the images of only two molecules.

► Page 13/14 The authors state "Furthermore, it is not without notice that the 2.1 nm periodicity of Type II G-wires is half of the 4.2 nm features of Type I" So, now what's the implication?

Answer: A possible explanation has been given on page 16.

► Figure 7 Where it states "(E-F) Rotamers of the slipped-strand model." It should be "(E-F) Rotamers of the (2,2) diagonal slipped-strand model."

Answer: Fig 7 and caption have been now updated.

► Either include references as evidence for the statement "Similar diversity at the stacking interface of G-quadruplex blocks has been previously reported" or take this out of the manuscript.

Answer: Reference has been now added for this.

► In Figure 8 "Atomic radii for Thymine and have been altered for visualization purposes." Correct to "...been altered"

Answer: This has been corrected.

Reviewers' Comments:

Reviewer #1 (Remarks to the Author):

The authors have addressed all my comments. On re-reading the manuscript and its figures, I just note two minor and easily addressable points at which it should be revised:

1) In Fig. 8f, the overlay of the structural model on the AFM data is insufficiently clear, which can probably be addressed by a change in colour scheme for the model or for the AFM data.

2) In the Methods, Atomic Force Microscopy (AFM) section and Fig. S22: The methods refer to ~5 pm thermal noise at ~100kHz. That does not make sense. Fig. S22 needs its axis labels to be made more readable, with the thermal noise given in pm/sqrt(Hz) versus frequency in kHz. And then it should be obvious that thermal noise at ~100 kHz should be stated in pm/sqrt(Hz), or in pm with a well-defined measurement bandwidth.

Reviewer #2 (Remarks to the Author):

The issues raised previously have been addressed

Responses to Reviewers:

Referee 1

► The authors have addressed all my comments. On re-reading the manuscript and its figures, I just note two minor and easily addressable points at which it should be revised:

1) In Fig. 8f, the overlay of the structural model on the AFM data is insufficiently clear, which can probably be addressed by a change in colour scheme for the model or for the AFM data.

Answer: We agree with the referee that the overlay of the model and the AFM data is not clear. We have decided to remove part (f) of the figure.

2) In the Methods, Atomic Force Microscopy (AFM) section and Fig. S22: The methods refer to ~5 pm thermal noise at ~100kHz. That does not make sense. Fig. S22 needs its axis labels to be made more readable, with the thermal noise given in pm/sqrt(Hz) versus frequency in kHz. And then it should be obvious that thermal noise at ~100 kHz should be stated in pm/sqrt(Hz), or in pm with a well-defined measurement bandwidth.

Answer: We thank the referee for pointing our mistake. We have now re-edited this section and removed Fig S22.